# Higher-Order Cumulants Drive Neuronal Activity Patterns, Inducing UP-DOWN States in Neural Populations

**DOI:** 10.3390/e22040477

**Published:** 2020-04-22

**Authors:** Roman Baravalle, Fernando Montani

**Affiliations:** 1Instituto de Física de La Plata (IFLP), Universidad Nacional de La Plata, CONICET CCT-La Plata, Diagonal 113 entre 63 y 64, La Plata, Buenos Aires 1900, Argentina; rbaravalle@fisica.unlp.edu.ar; 2Departamento de Física, Facultad de Ciencias Exactas, UNLP Calle 49 y 115. C.C. 67, La Plata, Buenos Aires 1900, Argentina

**Keywords:** neuronal activity, collective dynamics, information processing, PDF evaluation, dynamic states, higher-order correlations, Fisher information and Shannon entropy, critical fluctuations and binder cumulant, high activity states, low activity states

## Abstract

A major challenge in neuroscience is to understand the role of the higher-order correlations structure of neuronal populations. The dichotomized Gaussian model (DG) generates spike trains by means of thresholding a multivariate Gaussian random variable. The DG inputs are Gaussian distributed, and thus have no interactions beyond the second order in their inputs; however, they can induce higher-order correlations in the outputs. We propose a combination of analytical and numerical techniques to estimate higher-order, above the second, cumulants of the firing probability distributions. Our findings show that a large amount of pairwise interactions in the inputs can induce the system into two possible regimes, one with low activity (“DOWN state”) and another one with high activity (“UP state”), and the appearance of these states is due to a combination between the third- and fourth-order cumulant. This could be part of a mechanism that would help the neural code to upgrade specific information about the stimuli, motivating us to examine the behavior of the critical fluctuations through the Binder cumulant close to the critical point. We show, using the Binder cumulant, that higher-order correlations in the outputs generate a critical neural system that portrays a second-order phase transition.

## 1. Introduction

How synaptic input correlations from shared presynaptic neurons translate into membrane potential and spike-output correlations is still an open question [1,2,3,4,5,6,7,8]. Input correlations have been picked up in different brain regions at the level of the membrane potential when considering several neurons [9]. The output activity of a neuronal population depends typically on the input activity of a very large number of synaptic inputs. The interplay between inputs activity and the synchronized spiking in the outputs is not well known, and thus the structure underlying the correlated neuronal responses at population level is still not well deciphered [1,2,3,4,5,6,8].

The output spiking activity is usually associated with the outside world tasks like stimulus encoding, attention, cognition, perception and behavior [6,10,11,12,13,14,15,16,17]. It has been studied how the output spiking activity depends on the mean and variance of the inputs, but it remains unexplored whether these quantities might also affect higher-than-pairwise spike correlations in the outputs [1,2,3,18,19,20]. That is to say, it is still undisclosed what could be the exact amount of correlations carried by the moments higher than two in the joint distribution of firing. Thus, it is not yet well understood whether a distribution of inputs without skewness and kurtosis can also generate higher-order cumulants in the joint distribution of firing, and how they might affect the neural networks dynamics [1,2,3,18,19,20]. In particular, how the input statistics affects the third and fourth output moments.

Using an appropriate theoretical modeling of the firing patterns that realistically emulate a neuronal population ensemble, and its spiking correlated activity, constitutes an important step as it can provide important insight on the diverse interaction between structures of the brain. Different statistical approaches have been proposed to investigate the correlated spiking activity in the brain [2,3,4,5,15,16,17,18,19,20,21,22]. In particular the DG approach delivers a formalism that allows us to successfully model the correlated structure of the spiking neuronal activity of the cortex [2,3]. The DG model simulates the spiking activity of a neuronal ensemble by generating multivariate Gaussian samples that characterize the correlated inputs to the neuronal population and thresholding them [2,3]. The DG approach generates dense higher-order statistics and can emulate higher-order correlations observed in the cortex [2,3,23]. The model generates, through a network with a multivariate distribution, with a given mean *h* and covariance Λ using a fixed threshold, certain correlations in the outputs [2,3]. These correlations are usually characterized by the output firing rate, μ, and the covariance, Σ [2,3,23]. However, it remains still unknown whether moments higher than two in the output joint distribution of firing could also shape the dynamics of the neural populations simulated using the DG model [1,2,3,18,19,20].

In this paper, we carefully investigate how pure pairwise correlations in the inputs can generate correlations higher than two in the outputs, by exactly quantifying them for different scenarios of the mean and variance of the inputs. That is to say, we compute the spike correlation coefficients up to fourth order considering a population of neurons receiving pairwise correlated inputs within the DG model. We propose a combination of analytical and numerical techniques to estimate the first, second, third and fourth moments of the probability distribution of firing. We first perform numerical estimations of the Amari’s correlations coefficients up to fourth order for the neural simulations [8,21]. Secondly, we carry out an analytical and numerical expansion of the DG model to explore the impact of the estimation of the higher-than-second-order cumulants in the population dynamics. The main objective is to understand whether these higher-order components might provide further understanding of the neuronal dynamics when considering a population of finite size and in the asymptotic regime.

We examine the relationship between higher-order correlations and firing rate, and the behavior of the critical fluctuations in the outputs, through the fourth-order cumulant, considering different degrees of input correlations. Importantly, the dependence of the Binder cumulant as function of the correlation moments allows us to analyze higher-order fluctuations close to the critical point present in the model [24,25,26,27]. We investigate whether a large amount of pairwise correlations in the inputs may have any effect on quiescent (“DOWN state”) and highly active state (“UP state”) in the outputs [28,29,30,31,32]. In the next section we test the hypothesis that a large amount of pairwise correlations in the inputs affects higher-order cumulants of the joint distribution of firing and if this can induce “DOWN/UP states”. Through the estimation of the fourth-order cumulant and the related Binder cumulant [24,25,26,27], which is independent of the number of neurons at the transition point, we show that the DG model exhibit a second-order phase transition. The connection between the correlations of spiking activity and the Binder cumulant links two fundamental features of the neural code and critical states. Consequently, this provides us a measurable methodology to evaluate detectable proof for higher-order correlations and to assess its role in the dynamical properties of the neuronal systems.

## 2. Characterization of the DG Model Considering Spike Correlations Higher than Two

The spiking activity of the network was described by a random variable $X\in {\left\{0,1\right\}}^{n}$ with output mean μ and output covariance matrix Σ. Here *n* is the number of neurons in the network.

In the DG model [22,33], an *n*-dimensional latent random variable *U*, described by a thresholded Gaussian distribution with mean *h* and covariance matrix Λ, is the cause of the spiking activity of the network. If we call $\Omega_{i}$ the threshold for the *i*th component of *U*, we can define the random variable *X* as follows: *X* takes the value 1 if and only if $U_{i}>\Omega_{i}$, and 0 otherwise. In the following we take $\Omega_{i}=0\;\forall i$. Thus, the input of the network consists of a distribution with pairs interaction, defined through the parameters *h* and Λ, with a threshold in 0. It is this threshold that causes the higher-order correlations in the output of the network, so the activity of the network cannot be fully described by the output mean and covariance, μ and Σ. The relevance of DG model on the investigation of higher-order correlations has been previously pointed out in Refs. [2,3,23]. The main idea here is to provide a quantification of the higher-order cumulants of the probability distributions of firing. More precisely we decompose these higher orders, putting special emphasis on third- and fourth-order interactions, as they are also important to describe the neural networks dynamics [15,16,17].

First, we want to model the first two moments of the variable *X* through the two first moments of the variable *U*. Without loss of generality, we put $\Lambda_{ii}=1,\;i=1,\ldots ,n$. Next, we calculate the mean and the covariance of *X* by definition:(1)$$\begin{array}{cc} \mu_{i}=E\left[X_{i}\right] & =\sum_{l=0}^{1}l\;p\left(X_{i}=l\right)\\ \; & =p\left(X_{i}=1\right)=P\left(U_{i}>0\right)\\ \; & =1-P\left(U_{i}\leq 0\right)\\ \; & =1-\Phi \left(0\right)\\ \; & =\Phi \left(h_{i}\right), \end{array}$$
where Φ is the cumulative function of the univariate Gaussian, and $h_{i}$ is the *i*th component of the mean vector *h*. The last line comes from the fact that the distribution of $U_{i}$ is symmetric around the mean $h_{i}$; it is not a general result for all the input distributions. For the covariance elements we can make the same reasoning:(2)$$\begin{array}{cc} \Sigma_{ij} & =E\left[X_{i}\;X_{j}\right]-\mu_{i}\mu_{j}\\ \; & =\sum_{l=0,m=0}^{1}l\;m\;p\left(X_{i}=l,X_{j}=m\right)-\mu_{i}\mu_{j}\\ \; & =p\left(X_{i}=1,X_{j}=1\right)-\mu_{i}\mu_{j}\\ \; & =P\left(U_{i}>0,U_{j}>0\right)-\mu_{i}\mu_{j}\\ \; & =1-P\left(U_{i}\leq 0,U_{j}\leq 0\right)-\mu_{i}\mu_{j}\\ \; & =1-\Phi_{2}\left(0,0,\Lambda_{ij}\right)-\mu_{i}\mu_{j}\\ \; & =\Phi_{2}\left(h_{i},h_{j},\Lambda_{ij}\right)-\mu_{i}\mu_{j}. \end{array}$$

Here $\Phi_{2}$ is the cumulative function of the bivariate Gaussian with unit variances and correlation coefficient $\Lambda_{ij}$. The previous line is again valid only for symmetric distributions.

Summarizing, the connection between the input parameters *h*, Λ and the output parameters μ, Σ is given by the two equations: (3)$$\begin{array}{cc} \mu_{i} & =\Phi \left(h_{i}\right), \end{array}$$(4)$$\begin{array}{cc} \Sigma_{ij} & =\Phi_{2}\left(h_{i},h_{j},\Lambda_{ij}\right)-\Phi \left(h_{i}\right)\Phi \left(h_{j}\right), \end{array}$$
with the extra condition $\Lambda_{ii}=1$. This pair of equations have a unique solution for any admissible moments μ and Σ, and can be solved numerically [2,3].

Up to this point we only measure the first and second central moment of the spiking distribution. In order to quantify the third and fourth central moments, we must have a similar expression for the moments in terms of the input distribution parameters. The third central moment can be estimated as (see Appendix A for further details):(5)$$\begin{array}{cc} \Sigma_{ijk} & =E\left[\left(X_{i}-\mu_{i}\right)\left(X_{j}-\mu_{j}\right)\left(X_{k}-\mu_{k}\right)\right]\\ \; & =E\left[X_{i}X_{j}X_{k}\right]-\mu_{i}E\left[X_{j}X_{k}\right]-\mu_{j}E\left[X_{i}X_{k}\right]\\ \; & -\mu_{k}E\left[X_{i}X_{j}\right]+2\mu_{i}\mu_{j}\mu_{k}. \end{array}$$

The mean and the covariance of *X* can be described using Equations (3) and (4). The only term left to calculate is $E\left[X_{i}X_{j}X_{k}\right]$. Following the same reasoning as previously, we found that:(6)$$\begin{array}{cc} E\left[X_{i}X_{j}X_{k}\right] & =\sum_{l,m,n=0}^{1}l\;m\;n\;p\left(X_{i}=l,X_{j}=m,X_{k}=n\right)\\ \; & =p\left(X_{i}=1,X_{j}=1,X_{k}=1\right)\\ \; & =P\left(U_{i}>0,U_{j}>0,U_{k}>0\right)\\ \; & =1-\Phi_{3}\left(0,0,0,\Lambda_{ijk}\right)\\ \; & =\Phi_{3}\left(h_{i},h_{j},h_{k},\Lambda_{ijk}\right), \end{array}$$
where $\Phi_{3}$ is the cumulative density function of a trivariate Gaussian with a 3×3 covariance matrix $\Lambda_{ijk}$ corresponding to the random variables $X_{i}$, $X_{j}$ and $X_{k}$.

For the fourth central moment, the following result is found (see Appendix A for further details):(7)$$\begin{array}{cc} \Sigma_{ijkl} & =E\left[\left(X_{i}-\mu_{i}\right)\left(X_{j}-\mu_{j}\right)\left(X_{k}-\mu_{k}\right)\left(X_{l}-\mu_{l}\right)\right]\\ & =E\left[X_{i}X_{j}X_{k}X_{l}\right]-\mu_{i}E\left[X_{j}X_{k}X_{l}\right]\\ & -\mu_{j}E\left[X_{i}X_{k}X_{l}\right]-\mu_{k}E\left[X_{i}X_{j}X_{l}\right]\\ & -\mu_{l}E\left[X_{i}X_{j}X_{k}\right]+\mu_{i}\mu_{j}E\left[X_{k}X_{l}\right]\\ & +\mu_{i}\mu_{k}E\left[X_{j}X_{l}\right]+\mu_{j}\mu_{k}E\left[X_{i}X_{l}\right]\\ & +\mu_{i}\mu_{l}E\left[X_{j}X_{k}\right]+\mu_{j}\mu_{l}E\left[X_{i}X_{k}\right]\\ & +\mu_{k}\mu_{l}E\left[X_{i}X_{j}\right]-3\mu_{i}\mu_{j}\mu_{k}\mu_{l}. \end{array}$$

Again, the only term that must be calculated is $E\left[X_{i}X_{j}X_{k}X_{l}\right]$. As expected, this term is equal to $\Phi_{4}\left(h_{i},h_{j},h_{k},h_{l},\Lambda_{ijkl}\right)$, with $\Phi_{4}$ the cumulative function of the four-dimensional Gaussian, and $\Lambda_{ijkl}$ the 4×4 covariance matrix between the variables $X_{i}$, $X_{j}$, $X_{k}$ and $X_{l}$.

We can see that the third and fourth central moments depends also on the lower-order moments, so we may have a combination of interactions between four, three and two neurons. In order to make a splitting of interactions between one, two, three or four neurons, we appeal to the Information Geometry tools, and make a coordinate system transformation, to the so-called theta coordinates [33]. In the next section we link the DG approach with the Information Geometry coordinates to estimate the effect of the correlations higher than two in the firing probability distributions.

### 2.1. Coordinate Transformations for the Pool of Neurons

On the whole we can symbolize the neuronal population activity by a binary vector $x=(x_{1},\ldots ,x_{N})$ in the space *X* of all binary vectors of length *N*, where $x_{i}=0$ if neuron *i* is silent in some time window and $x_{i}=1$ if it is firing one or more spikes. The probability P(x) of observing a particular response can be portrayed using different coordinate systems. A useful way of characterizing the population activity distribution is by indicating the $2^{N}-1$ individual probability values; these are called the *p*-coordinates. The probability is identified by the $2^{N}-1$ marginals; usually known as the η-coordinates [21]. Supplied P(x)≠0 for a given x, any such distribution can be extended in the so-called log-linear model, or θ-coordinates system [21,22]:(8)$$\begin{array}{cc} P(x)= & \exp\left(\sum x_{i}+\theta_{i}+\sum_{i<j}x_{i}x_{j}\theta_{ij}+\right.\\ \; & +\sum_{i<j<k}x_{i}x_{j}x_{k}\theta_{ijk}+\cdots +\\ \; & \left.+\sum_{i<\cdots <N}x_{i}\cdots x_{N}\theta_{i\cdots N}-\psi \right) \end{array}$$
where the $2^{N}-1$ different θ coefficients exclusively resolve the above distribution. This coordinate system was developed by Amari and peers to investigate the possibilities and interactions between neurons [21,22].

Allow us to consider that the neural population is a totally homogeneous pool; all of the parameters portraying single neuron properties do not depend upon the accurate character of each neurons, but instead on the amount of neurons being considered. Due to the symmetry all the θ at a given order *i* are the same and can be rewritten by $\theta_{i}$. Within this framework, the probability of having exactly *m* neurons active at a certain time bin would be [8,20]:(9)P(m)∼Nmexp∑i=1Nmiθi−ψ.

That is, we assume that all the parameters characterizing single neuron properties and interactions between any group of neurons do not depend on the precise identity of the considered neurons, but rather on the number of neurons. Let us consider therefore a pool of *N* neurons where each unit has a membrane potential $u_{i}$ subject to a joint Normal distribution. Given U∼N(h,Λ), where $\Lambda =I\left(1-\alpha \right)+\alpha 1_{N}1_{N}^{T}$, then
(10)$$u_{i}=h+\sqrt{1-\alpha }v_{i}+\sqrt{\alpha }\varepsilon .$$

The variables $v_{i}$ and ε are two independent random variables subject to the Gaussian distribution N(0,1) with mean 0 and variance 1. The input statistics are chosen such that the outputs *X* have mean μ and covariance Σ. Since here we focus on the case of homogeneous populations then $\mu_{i}=\mu$ and $\Sigma_{ij}=\sigma$, and we name the pairwise correlation coefficient as
(11)$$\rho =\frac{\sigma }{\mu (1-\mu )}.$$
the skewness in Equation (5) as $\zeta =\Sigma_{ijk}$ and the kurtosis coefficient in Equation (7) as $\chi =\Sigma_{ijkl}$.

### 2.2. Synchronized Activity of the Outputs Considering a Neuronal Pool

Several investigations pointed out the significance of higher-order correlations, as pairwise models fail to demonstrate the variability of the neuronal ensembles at a global level [8,14,15,16,17,18,34,35,36]. In the DG model, designed to produce spiking patterns thresholding a multivariate random Gaussian variable, spike correlations across neurons emerge from interactions in the inputs [2,3,33]. They are usually quantified through the mean firing rate as in Equation (3) and by second-order cumulant depicted in Equation (4) [2,3,33]. Next, let us investigate how input correlations may influence the spiking correlated activity of the outputs. Along the following lines we portray the DG model estimating the first-, second-, third- and fourth-order θ coefficients of the Amari’s expansion [21,22] when considering different input correlations $\alpha ={\left(\Lambda \right)}_{ij},i\neq j$. We consider a neuronal population of n=50 neurons simulated using the DG model [2,3,33]. Figure 1 shows how $\theta_{2}$ grows non-linearly as the mean firing rate μ increases, for different values of the input correlations α. Figure 2 shows $\theta_{3}$ versus μ that present the opposite behavior to Figure 1 as in this case the third order decreases non-linearly with the mean firing rate μ, for different values of the input correlations α. Please note that $\theta_{3}$ versus μ takes larger negative values as μ increases. In addition, Figure 3 depicts the fourth-order correlation coordinate $\theta_{4}$ for different values of the input correlations α, which takes negative values for very low firing rates, μ, but grows positively and non-linearly as μ becomes bigger. Overall this probabilistic model is imposing non-trivial constraints on population-level statistics as it appears that the third-order term helps to inhibit the correlation contribution to the global distribution of firing, because $\theta_{3}$ is mostly a negative correlation coefficient (see Figure 2). We show using the Amari’s formalism that the third term gives a solid restraint to the general DG system action, while the fourth-order term represents a sort of excitatory balance in the probability distribution. That is $\theta_{4}$ is mostly a positive correlation coefficient (see Figure 3).

To detect the dynamical changes in the probability distributions we estimate Fisher information and Shannon entropy. Let us emphasize that Fisher information can be deciphered as an extent of the capacity to assess a parameter and as the amount of information that can be extracted from a set of measurements [37,38], while entropy is a measure of uncertainty [39]. We refer the reader to Appendix B for the detailed definitions. We show in Figure 4 and Figure 5 that the effects of inputs correlations are not negligible in the output spiking activity when estimating the Shannon entropy and Fisher information, respectively. Both Shannon entropy and Fisher information depict an opposite behavior as the firing rate grows, for different values of the input correlations α. Notice that entropy becomes bigger as function of μ when the input correlation α is smaller. In contrast Fisher information is smaller as the firing rate μ becomes bigger for lower values of α. Thus, Fisher information is larger as the variance of the inputs grows. The previous results show that the effect of correlations in the inputs induces dynamical changes in the probability distributions that are significant on information transmission in the outputs. Moreover, as is well known, when describing the behavior of Fisher information close to a critical regime this quantity grows as the system is near a phase transition [40,41]. In the next sections we investigate the emergent properties of this system, the possible link between information theory and thermodynamic interpretations of critical behavior.

Despite the previous evidence showing non-negligible effects of correlations higher than two, we cannot interpret from the analysis depicted above whether they might reflect the mechanisms responsible for tuning the network of stochastic neurons towards a critical boundary. Spiking correlations have proven to be important for the characterization of the DG model, which are characterized by the estimations of the cumulants depicted in Equations (3) and (4) [2,3,33]. This model has been broadly used to build quantitative forecasts on how takeoffs from pairwise models rely upon normal Gaussian like neuronal inputs and to create an ensemble of neurons with a spike trains with indicated mean and pairwise statistics [2,3,33]. However, we consider that the effect of the higher-order cumulants also must be taken into account [2,3,33], and the main idea here is to gain further understanding of the neuronal dynamics by extending this approach to higher orders. Thus, in the following section, we perform an extended analysis of the DG model to gain a broader picture of the role of higher correlations on the neural network dynamics.

### 2.3. Higher-Order Cumulants and the Identification of a Second-Order Phase Transition

The DG model mimics population spiking action by producing multivariate Gaussian examples and thresholding them [2,3,33]. Dissimilar to most of the maximum entropy models [12,13,42], we can legitimately indicate the firing rates for the DG model so as to create cortical-like measurements [2,3,23]. It has been shown that correlations are important for the characterization of the DG model as it generates dense higher-order statistics and can replicate higher-order spike correlations [2,3,33]. It is our goal to show that a proper quantification of the third and fourth moment of the spike-count distribution is useful for characterizing and, in some cases, anticipating a change in the spiking behavior of the population. This means that the skewness and kurtosis could be important, not only in the theoretical case, but also in the characterization of neuronal ensembles in the cerebral cortex. Let us consider that the neural array is a totally homogeneous pool; all of the parameters depicting single neuron properties and interactions between any ensemble of neurons do not depend on the precise identity of the considered neurons, merely just on the amount of neurons considered. In the current simulations we also consider a population of 50 neurons within the DG model. In order to perform a proper characterization of the second, third and fourth moment of the spike-count distribution in the outputs we take several values for the covariance in the inputs. Figure 6 shows the pairwise correlation coefficient ρ versus the mean firing rate μ. This figure demonstrates that this model has a remarkable connection between pairwise correlations ρ and the firing rate μ, which is likely to be found in neural ensembles, and depicts that the pairwise correlation coefficient ρ increases monotonically with firing rate up to μ=0.5. Figure 6 depicts a maximum at μ=0.5 and we have selected various values of the input correlations α=0.4, α=0.45, α=0.5, α=0.55, α=0.6, α=0.7, α=0.8, α=0.9 and α=0.95. Figure 7 shows the skewness ζ as a function of the mean firing rate μ of the neuronal population, when considering several values of the input correlations α=0.4, α=0.45, α=0.5, α=0.55, α=0.6, α=0.7, α=0.8, α=0.9 and α=0.95. Note from Figure 7 that the skewness changes its sign at μ=0.5, and presents a maximum for low firing rates and a minimum when considering very high firing rates.

Figure 8a shows the kurtosis χ versus the mean firing rate for different values of the input correlations α=0.4, α=0.45, α=0.5, α=0.55, α=0.6, α=0.7, α=0.8, α=0.9 and α=0.95. Notice from Figure 8a that there is a change in the shape of kurtosis for values of the input correlation α bigger than 0.7. Please note that the kurtosis depicts a local minimum at μ=0.5 when considering values of the input correlation α bigger than 0.7. Bottom Figure 8b represents a zoom of top Figure 8a, and shows that as α becomes equal to 0.8, 0.9 and 0.95 the kurtosis of the distribution depicts a bimodal behavior. Interestingly, below α=0.7 the kurtosis is unimodal but for higher values the kurtosis show a bimodal behavior with the two peaks that indicates a broken-symmetry due to a induced phase transition, i.e., the fourth-order cumulant becomes highly suggestive that long-range fluctuations appear for a critical values of α between 0.8 and 0.95. Below these critical values of α, the maximum of kurtosis occurs at μ=0.5 for which the kurtosis shows a single peak. The previous results are similar for different mean *h* in the inputs.

Let us note that for very low values of μ and pairwise interactions in the inputs bigger than α=0.7, Figure 7 and Figure 8 depict a maximum with positive skewness and kurtosis. On the contrary taking a firing rate μ close to the maximum value and pairwise interactions in the inputs bigger than α=0.7 leads to a minimum (negative) skewness and positive (maximum) kurtosis in the outputs (see Figure 7 and Figure 8). The bimodal behavior for the kurtosis seems to indicate that two states with maximum kurtosis that are accompanied with maximum or minimum skewness may coexist. This suggests that a phase transition could take place and that the third- and fourth-order cumulants might be of help to characterize it. Next, in order to gain further understanding of the nature of those cumulants we estimate them in the asymptotic limit.

#### The Asymptotic Behavior

To calculate the higher-order cumulants in the limit of an infinite number of neurons, let us consider a model of a homogeneous population with a number *n* of neurons [3]. Each neuron has a firing rate μ and constant pairwise covariance ρ. We obtain samples *X* from the model by dichotomizing a latent Gaussian *U* with mean *h*, unit variances and pairwise covariance (and correlation coefficient) α>0 [2,3]. We decompose the Gaussian *U* into three parts: the constant corresponding to the mean, one multidimensional independent Gaussian *V* with zero mean and a univariate Gaussian distribution ε with zero mean and variance equal to 1:$$\begin{array}{cc} U∼ & N\left(h,\Lambda \right)\mathrm{where}\Lambda =I_{n}\left(1-\alpha \right)+\alpha 1_{n}1_{n}^{T},\\ U= & \;h+\sqrt{1-\alpha }V+\sqrt{\alpha }\varepsilon 1_{n},\\ \mathrm{where} & V∼N(0,I_{n})\\ \mathrm{and} & \varepsilon ∼N(0,1). \end{array}$$
$1_{n}$ is the unitary vector, formed by all ones. Such pool of neurons is fully characterized by the spike-count distribution of the population [2,3,33]. We name *k* as the spike count in the population, and r=k/n the proportion of spiking neurons. The probability of having *k* spikes is QDG(k)=PDG(k)/nk [2,3]. Hence, the population spike-count distribution is
$$\begin{array}{cc} P_{DG}\left(X=x\right) & =\int_{u}P\left(X=x|u\right)p\left(u\right)dz\\ \; & =\int_{-\infty }^{\infty }\phi_{\alpha }\left(\varepsilon \right)P\left(X=x|\varepsilon \right)d\varepsilon \\ \; & =\int_{-\infty }^{\infty }\phi_{\alpha }\left(\varepsilon \right)\prod_{i=1}^{n}P\left(X_{i}=x|\varepsilon \right)d\varepsilon \\ \; & =\int_{-\infty }^{\infty }\phi_{\alpha }\left(\varepsilon \right)\prod_{i=1}^{n}{\left(1-L\left(\varepsilon \right)\right)}^{1-x_{i}}L{\left(\varepsilon \right)}^{x_{i}}d\varepsilon \\ \; & =\int_{-\infty }^{\infty }\phi_{\alpha }\left(\varepsilon \right){\left(1-L\left(\varepsilon \right)\right)}^{n-k}L{\left(\varepsilon \right)}^{k}d\varepsilon \\ \; & =\int_{-\infty }^{\infty }\phi_{\alpha }\left(\varepsilon \right)\exp\left(n\;W(s)\right)d\varepsilon \\ \mathrm{where}W(\varepsilon ) & =r\log L(\varepsilon )+(1-r)\log L(-\varepsilon )\\ \mathrm{with}L(\varepsilon ) & =P(u_{i}>0|\varepsilon )\\ \; & =P\left(V_{i}>-\frac{\varepsilon +h}{\sqrt{1-\alpha }}\right)\\ \; & =\Phi \left(\frac{\varepsilon +h}{\sqrt{1-\alpha }}\right). \end{array}$$

Here $\phi_{\alpha }\left(\varepsilon \right)$ is the probability density function of a one-dimensional Gaussian with mean equal to 0 and variance α [2,3]. Φ is the cumulative function of a standard one-dimensional Gaussian.

For large population sizes *n*, we can use the saddle-point approximation in the last integral. After some calculations one can obtain the asymptotic continuous valued spike-count distribution f(r), as in Ref. [2,3]:$$\begin{array}{cc} f(r)= & \sqrt{\frac{1-\alpha }{\alpha }}\exp\left(\frac{h^{2}}{2-4\alpha }\right)\\ \; & \exp\left(-\frac{\left(1-2\alpha \right){\left(\Phi^{-1}\left(r\right)-\frac{h\sqrt{1-\alpha }}{1-2\alpha }\right)}^{2}}{2\alpha }\right). \end{array}$$

In the following we use f(r) to estimate the asymptotic pairwise cumulant ρ, the skewness ζ and the kurtosis χ of the distribution.

Figure 9a–c depict the asymptotic estimations of the second-, third- and fourth-order cumulants, respectively. The asymptotic limit has a similar behavior as the previous DG simulation considering a finite number of neurons. The current findings emphasize the results of the previous section. In the following we investigate how higher-order correlations in the outputs might induce neural criticality. The partition-function has a discontinuity at α=1/2. Thus, the transition point is at α=1/2, independently of the mean *h* of the latent Gaussian. For values α<1/2 the distribution is unimodal, while for α>1/2 the distribution is bimodal. Next, let us consider the heat capacity and the fourth-order cumulant dependence with temperature. To do it so we derive, using the ideas in Ref. [2], the behavior of the specific heat and the Binder cumulant as a function of temperature. Given the model distribution P(x), the extension at any temperature T=1/β is $P_{\beta }\left(x\right)=P{\left(x\right)}^{\beta }/Z_{\beta }$. Then, the spike-count distribution is given by $P_{\beta }\left(k\right)=n\exp\left(n(1-\beta )\eta (r)\right)\frac{P{\left(k\right)}^{\beta }}{Z_{\beta }}$, where η(r)=−rlog(r)−(1−r)log(1−r) [2,3].

It is useful to calculate the entropy rate $h_{DG}=\frac{1}{n}H\left(x\right)$ of the variable *X*. As shown in Ref. [2,3], for large populations, the result is:$$\begin{array}{c} h_{DG}\approx \int_{0}^{1}f_{\beta }\left(r\right)\eta \left(r\right)dr, \end{array}$$
where the subscript β indicates the dependence with the temperature. Taking into account that the specific heat is $c=Var\left[\log\left(P_{\beta }\left(x\right)\right)\right]/n$, and that we are in the limit of large *n*, the dominant term in *c* is [2,3]:$$\begin{array}{c} c\approx n\int_{0}^{1}f_{\beta }\left(r\right){\left(\eta \left(r\right)-h_{DG}\right)}^{2}dr. \end{array}$$

Extending the results in Refs. [2,3], we calculate the fourth-order cumulant. We depict here step by step the calculation. First, we define the fourth-order cumulant $\nu_{4}=\sum_{x}P_{\beta }\left(x\right){\left(\log\left(P_{\beta }\left(x\right)\right)-E\left[\log\left(P_{\beta }\left(x\right)\right)\right]\right)}^{4}$. Using the fact that the sum over *x* could be replaced by a sum over *k* (due to the fact that the neurons are identical), ∑x=∑knk, and that Pβ(x)=Pβ(k)/nk≈fβ(r)n/nk, the fourth-order cumulant is:ν4≈∑kfβ(r)nlogfβ(r)n−lognk+Hn4.

Defining nη(r)=nk=nnr, using the Stirling approximation for the binomial coefficient, and approximating $\frac{\sum_{k}}{n}\approx \int_{0}^{1}dr$, we arrive at:$$\begin{array}{cc} \nu_{4} & \approx \int_{0}^{1}f_{\beta }\left(r\right){\left(\log f\left(r\right)-\log n-n\eta \left(r\right)+nh_{DG}\right)}^{4}\;dr\\ \; & =\int_{0}^{1}f_{\beta }\left(r\right){\left(\log\frac{f(r)}{n}-n\left(\eta \left(r\right)-h_{DG}\right)\right)}^{4}dr. \end{array}$$

As in the limit of large populations the dominant term is the one which is linear with *n* inside the parentheses, thus the final result is:(12)$$\begin{array}{c} \nu_{4}=n^{4}\int_{0}^{1}f_{\beta }\left(r\right){\left(\eta \left(r\right)-h_{DG}\right)}^{4}dr. \end{array}$$

Considering that the second-order cumulant can be written as $\nu_{2}=Var\left[\log\left(P_{\beta }\left(x\right)\right)\right]=n\;c$ and that the fourth-order cumulant is given by Equation (12), we define the Binder Cumulant as [24,25,26,27]:$$\begin{array}{c} B=1-\frac{\nu_{4}}{3\nu_{2}^{2}}. \end{array}$$

Based on these estimations we can explore, as in Ref. [3], the scaling properties of the population in function of the temperature. In this case, the fourth-order moment is also important, because it allows us to calculate the Binder cumulant [24,25,26,27]. This cumulant is a robust way in which one can obtain the critical temperature of a system, finding the crossing points between this function for various population sizes [24,25,26,27]. In Figure 10 we depict the Binder cumulant as function of the temperature, and it is shown, as expected, that the critical point lies at T=1. This cumulant is less biased in temperature than the specific heat, shown in Figure 11, which makes it a very good quantifier to perform finite size scaling analysis for a neuronal system depicting a critical phase transition. Thus, by computing the Binder critical cumulant which does not depend on the quantity of neurons at the transition point, we show that the system displays a second-order phase transition. Similar results are obtained for the Binder cumulant using the numerical estimation of the second- and fourth-order cumulant presented above. The current finding emphasize the relevance of estimating the higher-order terms, as the expanded moments of the proposed models capture the dynamics of the interconnected network in neural spiking through the estimation of the Binder cumulant. We show through the estimation of the higher-order cumulants the existence of a phase transition of second order. Our analysis reveals that the coding of the higher than second-order moments could dominate the brain dynamics as the stochastic restriction of the neurons leads them towards a critical boundary.

## 3. Discussion and Conclusions

The principal notable property of neural ensembles is that they are globally coupled. While it has been known for quite a while that neurons do not spike autonomously, early estimations of their cooperative contributions of functional connectivity were initially portrayed using pairwise correlations models (Ising-like) [12,13,42]. In particular the Ising model in the framework of the maximum entropy principle has been associated with a certain correlation between pairs of neurons in the retina [12,13,42]. As the technology made possible to simultaneously record the activity of hundred neurons in the cerebral cortex, be that as it may, it turned out to be progressively clear that pairwise models are inadequate to completely capture statistics in different data [8,14,15,16,17]. Moreover, it has been demonstrated that these intricate correlations are progressively adjusted by the stimulus [16] and that they can affect coding properties [8].

On the one hand it has been proposed that instead of directly estimating particular correlation parameters, it is possible to calculate the cumulant-based inference of higher-order correlations (CuBIC) to derive a lower bound in a given neuronal data set on the order of correlation [4,5]. On the other hand it has been shown that when modeling the collective behavior of biological systems, arisen from real data, the factual knowledge about the system is balanced close to an extremely uncommon point in their parameter space: a critical point [43]. Changes in triplet connections can altogether upgrade coding if those correlations depend on the stimuli, and it is through the skew that triplet correlations induce changes in population-wide distributions of neural responses [34]. Moreover, recent findings have shown the relevance of the kurtosis in the distribution of firing [20], and it has been also shown that the kurtosis is a very good quantifier for the description of the scaling properties of a neural population [44]. Higher correlations are indeed important for understanding neuronal avalanches [23] and even synchronous silence, or the co-inactivation of neurons, which is an omnipresent evidence of higher-order correlations [35]. Thus, assessing the accurate measure of pairwise, triplewise and quadruplewise correlations can be of great assistance to describe important functional properties of the cortex.

Cortical type models such as the DG have been used to generate spike trains with certain firing rates and degrees of correlation between action potentials [2,3,23,33]. For this purpose, it has been evaluated how different means and variances in the DG model inputs can affect the statistics of the neuronal firings in the outputs [2,3,23,33], and how statistical deformations of the Gaussian inputs changes the dynamics of the outputs [19,20,45]. In particular, a recent paper has shown the importance of maximum entropy models for analyzing the firing patterns of neuronal populations and the level of correlation between them [46]. In order to investigate the role of higher-order correlations in the neural code, we performed in this paper an extension of DG model proposed in Refs. [2,3] quantifying the contributions of the higher-order cumulants. Our current approach allows us to infer that, when considering strongly correlated inputs, a balance between skewness and kurtosis can induce jumps between “DOWN and UP states” in neuronal a population and this corresponds to a second-order phase transition, i.e., thought this approach we describe the basis of the dynamic mechanism by which these states transition occur.

In this paper, we show first using the Amari’s formalism on the DG model that the third-order term provides a strong inhibition to the overall DG networks activity, while the fourth-order term account for excitatory contributions in the probability distribution providing insight that higher-order correlations should not be disregarded. Thereafter we extended the DG model [2,3,23,33] and to implement this we estimate the skewness and kurtosis of the simulated population while broadly changing the input parameters of the population, i.e., the inputs correlations of the Gaussian distribution. In the current study we take into account all correlations orders as it is important to consider that the neuronal systems work close a critical point to improve information transmission in the cortex [47,48]. The extended DG model permits us to emulate the activity of the neuronal cortex but it is through higher-order cumulants that the estimation of the Binder cumulant allows us to provide evidence showing that this realistic neuronal network exhibit a second-order phase transition. More importantly, we show using the extended DG model that higher-order correlations are the mechanisms responsible for tuning the neuronal network towards the critical state. Our findings show on one hand that high pairwise interactions in the inputs could be sufficient to generate a third- and fourth-order cumulant in which a positive skewness and kurtosis can induce the system to a “DOWN state” (quiescent) [28,29,30,31,32]. On the other hand high pairwise interactions in the inputs can produce third- and fourth-order cumulant in which a negative skewness and positive kurtosis can induce the system to an “UP state” (active) [28,29,30,31,32]. Thus, if the amount of pairwise correlations in the inputs is high enough there is always a nonzero probability that the system could pass from a quiescent to a highly active state, by inducing a change in the sign of the skewness and taking advantage of the bimodal shape of the kurtosis. Near the critical point the system can be maximally responsive, and this might be therefore a mechanism to optimally managing information about the external stimuli. Indeed, the reduced fourth-order cumulant (or Binder cumulant) allows us to show that a bimodal shape with the two peaks corresponds to the critical behavior of a second-order phase transition [24,25,26,27].

Numerous brain functions are optimized due to the fact that the brain operates between two coexisting phases of order and disorder [49]. The nervous system operates near a non-equilibrium phase transition where a large amount of spatial and temporal higher-order correlations are compatible. The brain keeps itself at the border of a whole instability urging a great number of adaptations at different scales which resemble a critical state. Criticality appears as a fundamental element of cortical boosts, and higher-order synchrony of firing neurons that are needed to investigate the emergent properties in spatial structure of the cortical activity and its reorganization become power laws at this transition point [23,49]. In the current paper we were able to investigate the collective dynamics of neuronal activity developing an extension of the DG model that quantifies the contributions of the higher-order cumulants, allowing us to investigate the scaling properties of the system. In addition, by combining different techniques that goes from Information Geometry to a DG extended approach we investigate higher interactions among a neuronal network. Our current study allows us to identify correlations of order higher than two as key factors that determine phase transitions in the DG model, considering that at the level of the Amari’s coordinates the triplet correlations gives an inhibitory-like restraint to the general DG system while the fourth-order term represents excitatory-like contributions depicting the non-locality of neural populations. However, it is the change of sign of skewness for high values of pairwise inputs correlations and the balance with a positive kurtosis what determines the possible excursions from quiescent to highly active states.

The DG model emulates dense higher-order statistics and can simulate higher-order correlations that can be recorded in vivo within specific areas of brain cortex [2,3,23]. In contrast with maximum entropy models [12,13,42], we can choose the firing rate in the DG model to emulate a cortical statistic patterns and that produces the same pairwise, triplewise and quadruplewise cumulant being quantified by ρ, ζ and χ, respectively. Using the DG simulation, we find a non-monotonic relationship between output higher-order correlations and firing rate, motivating us to investigate the behavior of the critical fluctuations by means of the Binder cumulant. Thus, we consider that delivering an appropriate framework for estimating the exact amount of correlations that pair, triple, and quadruple moments are carrying about the joint firing distribution in the DG model is of ultimate relevance for understanding further the emergent properties of cortical processing. Importantly, our discoveries demonstrates that a high amount of pairwise correlations in the inputs can yield to a third and a fourth cumulant that leads to a framework with to two different scenarios, one with very low firing rate (“DOWN state”) and another with very high firing rate (“UP state”). Thus, the extended DG tools can be used to investigate further the complexity of different dynamic responses within neural populations of the cerebral cortex.

## Figures and Tables

**Figure 1 entropy-22-00477-f001:**
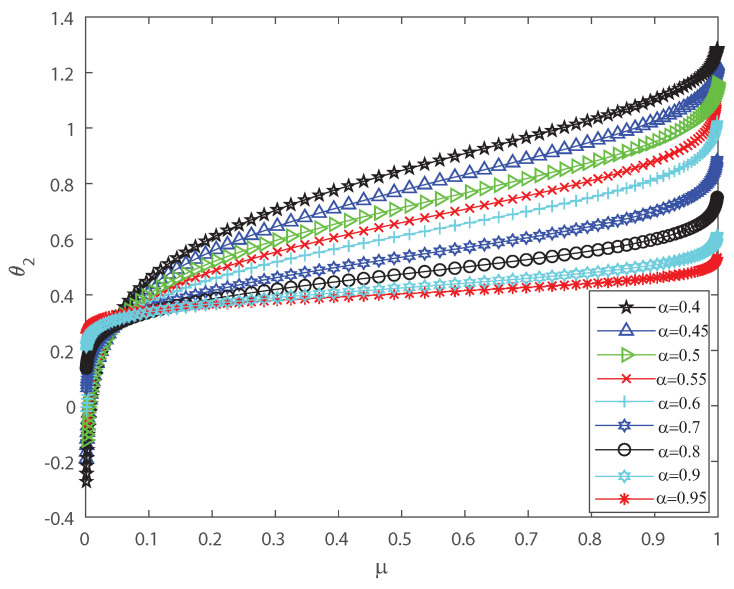
$\theta_{2}$ coordinate versus the mean firing rate μ, for different values of the input correlations α=0.4, α=0.45, α=0.5, α=0.5, α=0.55, α=0.6, α=0.7, α=0.8, α=0.9 and α=0.95. $\theta_{2}$ grows non-linearly as the firing rate μ increases. We consider n=50.

**Figure 2 entropy-22-00477-f002:**
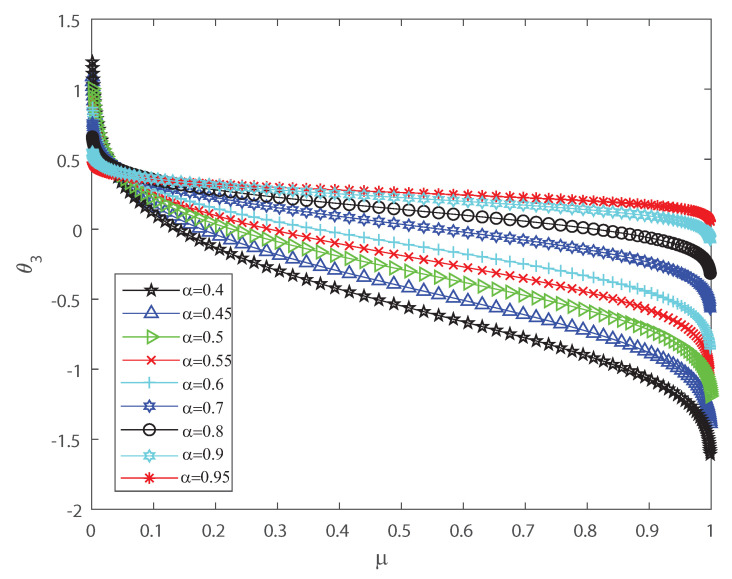
$\theta_{3}$ coordinate versus the mean firing rate μ, for different values of the input correlations α=0.4, α=0.45, α=0.5, α=0.5, α=0.55, α=0.6, α=0.7, α=0.8, α=0.9 and α=0.95. $\theta_{3}$ decreases non-linearly as the firing rate μ increases. We consider n=50.

**Figure 3 entropy-22-00477-f003:**
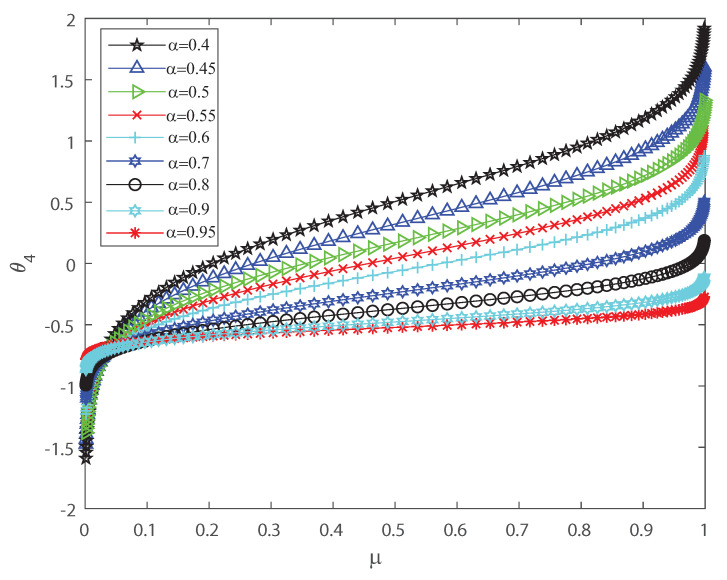
$\theta_{4}$ coordinate versus the mean firing rate μ, for different values of the input correlations α=0.4, α=0.45, α=0.5, α=0.5, α=0.55, α=0.6, α=0.7, α=0.8, α=0.9 and α=0.95. $\theta_{4}$ grows non-linearly as the firing rate μ increases. We consider n=50.

**Figure 4 entropy-22-00477-f004:**
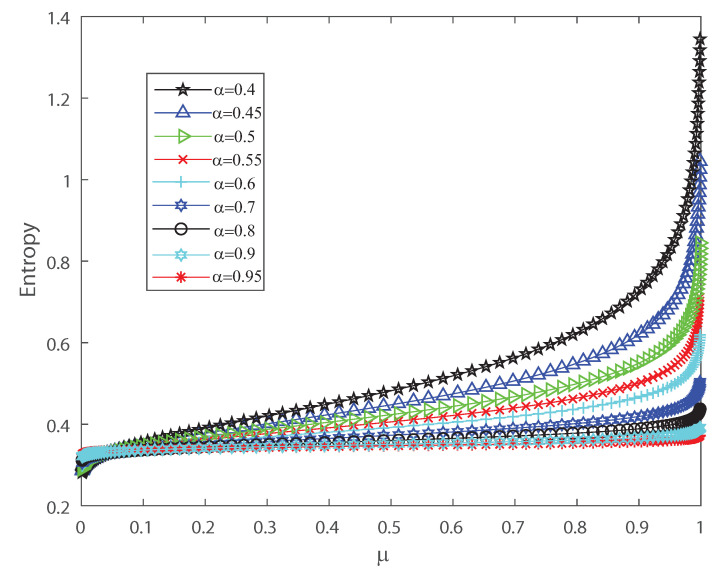
Shannon entropy versus the mean firing rate μ, for different values of the input correlations α=0.4, α=0.45, α=0.5, α=0.5, α=0.55, α=0.6, α=0.7, α=0.8, α=0.9 and α=0.95. We consider n=50.

**Figure 5 entropy-22-00477-f005:**
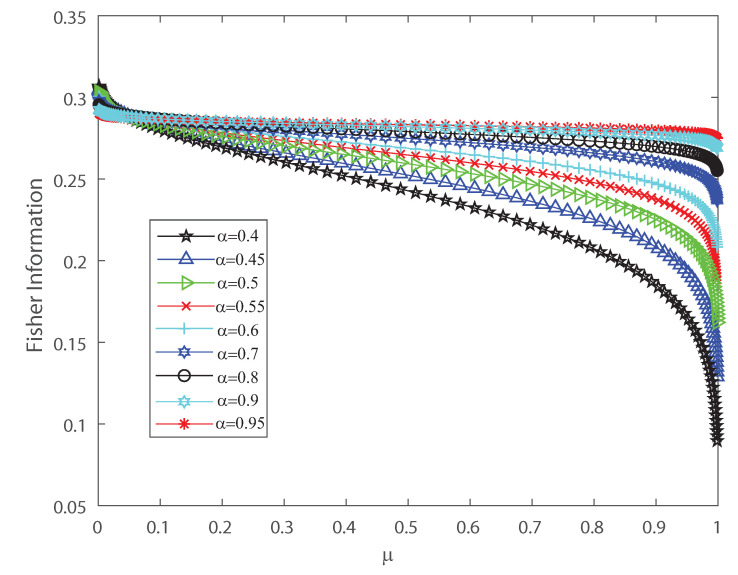
Fisher information versus the mean firing rate μ, for different values of the input correlations α=0.4, α=0.45, α=0.5, α=0.5, α=0.55, α=0.6, α=0.7, α=0.8, α=0.9 and α=0.95. We consider n=50.

**Figure 6 entropy-22-00477-f006:**
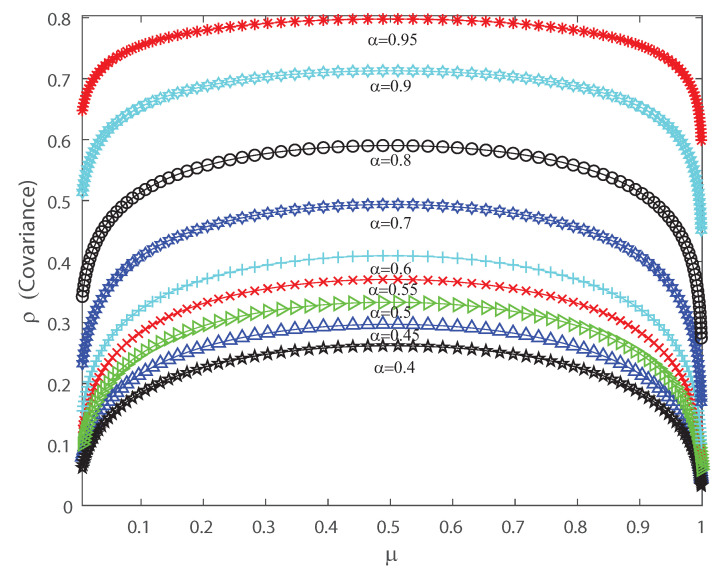
Covariance ρ as function of the population mean firing rate μ for various values of the input correlations α=0.4, α=0.45, α=0.5, α=0.5, α=0.55, α=0.6, α=0.7, α=0.8, α=0.9 and α=0.95. We consider n=50.

**Figure 7 entropy-22-00477-f007:**
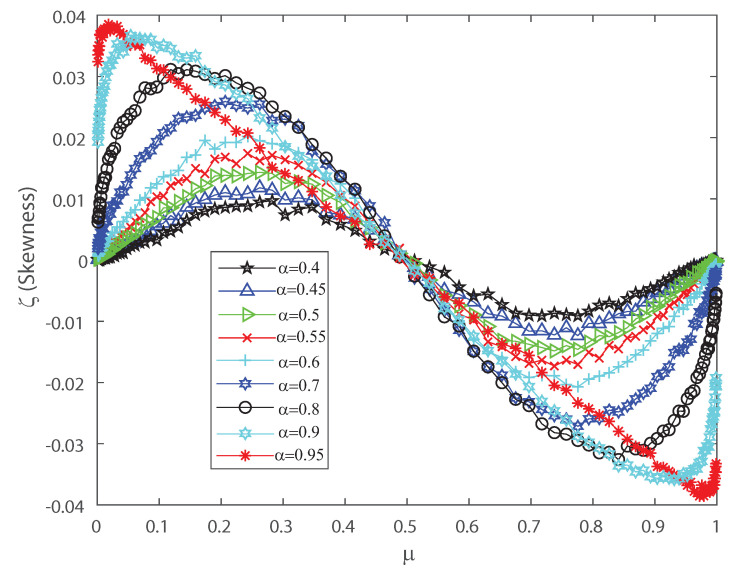
Skewness ζ as function of the population mean firing rate μ for various values of the input correlations α=0.4, α=0.45, α=0.5, α=0.5, α=0.55, α=0.6, α=0.7, α=0.8, α=0.9 and α=0.95. We consider n=50.

**Figure 8 entropy-22-00477-f008:**
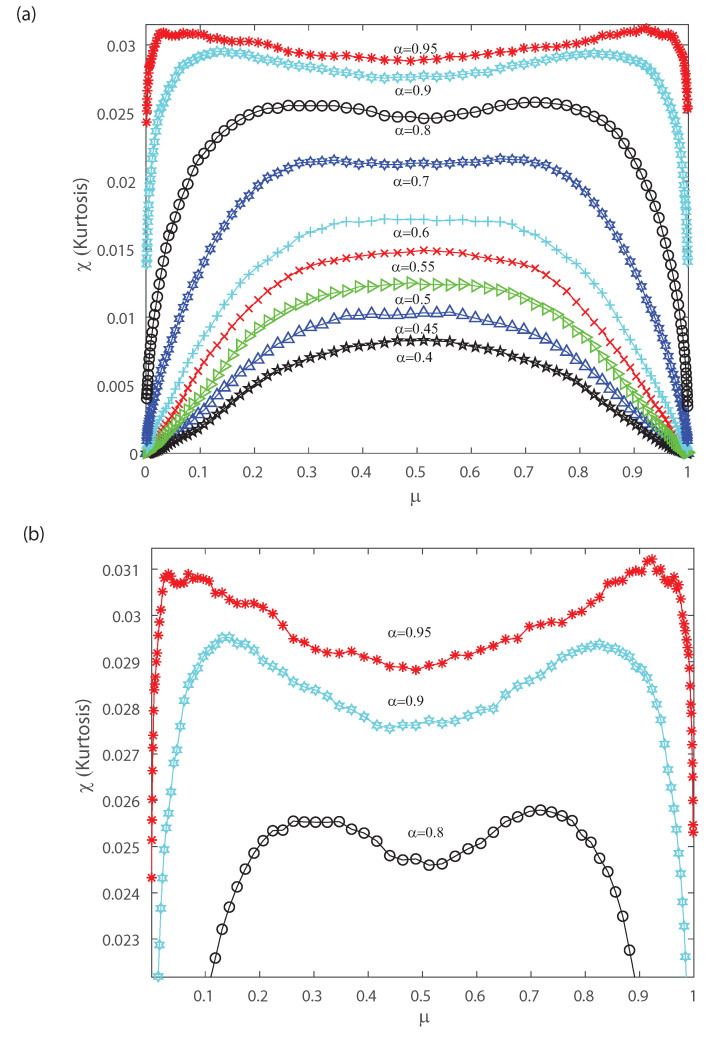
Kurtosis χ as a function of the population mean firing rate μ. (**a**) We consider various values of the input correlations α=0.4, α=0.45, α=0.5, α=0.5, α=0.55, α=0.6, α=0.7, α=0.8, α=0.9 and α=0.95. (**b**) Zoom of the top figure for α=0.8, α=0.9 and α=0.95. We consider n=50.

**Figure 9 entropy-22-00477-f009:**
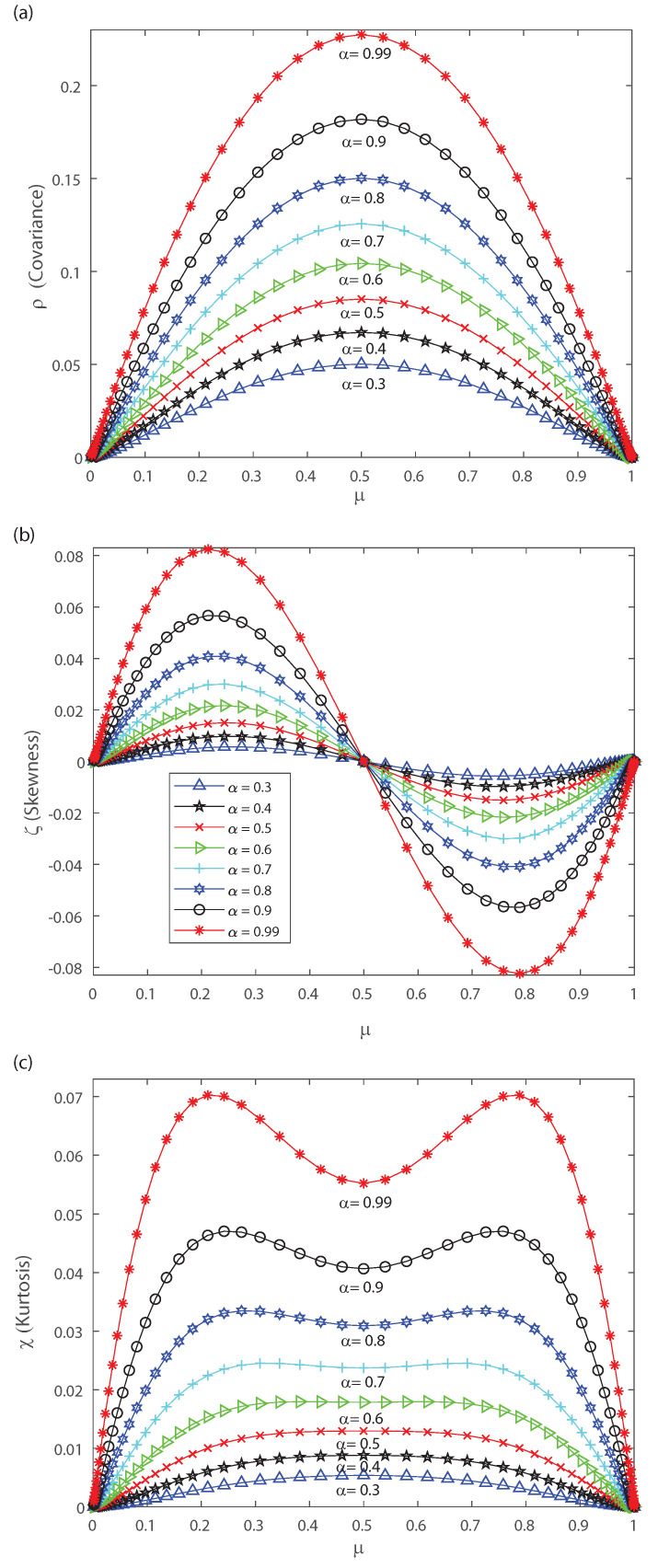
Asymptotic estimations of the DG cumulants. We consider several values of the input correlations α=0.3, α=0.4, α=0.5, α=0.6, α=0.7, α=0.8, α=0.9 and α=0.99. (**a**) Covariance ρ versus the mean firing rate μ. (**b**) Skewness ζ versus the mean firing rate μ. (**c**) Kurtosis χ as a function of the population mean firing rate μ.

**Figure 10 entropy-22-00477-f010:**
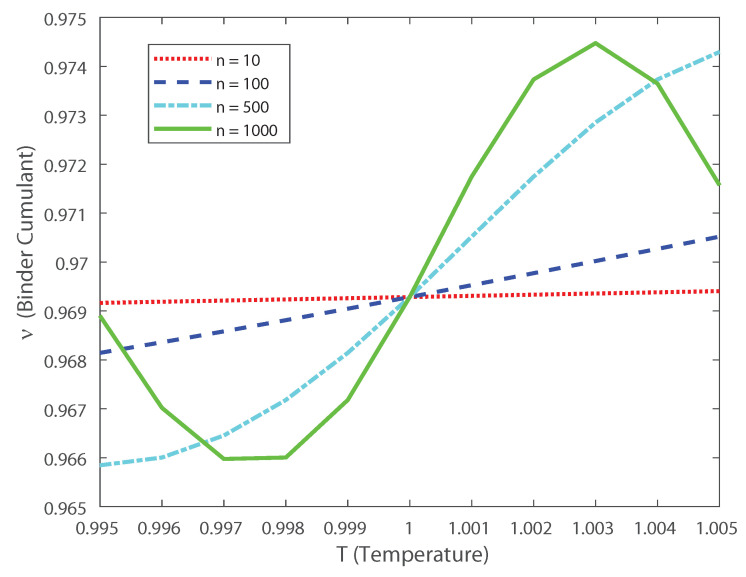
Binder cumulant as a function of the temperature for various values of the population size *n*. Notice that even for low values of populations (e.g., n=10) the crossing point between the curves lies at the critical temperature T=1.

**Figure 11 entropy-22-00477-f011:**
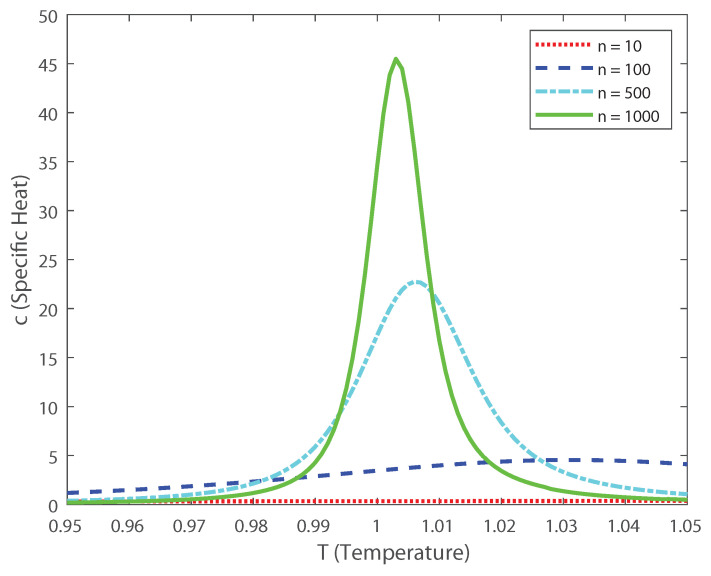
Specific heat as a function of the temperature for various values of the population size *n*.

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
