# Peer review of "Higher-Order Cumulants Drive Neuronal Activity Patterns, Inducing UP-DOWN States in Neural Populations"

_entropy, 2020, doi:10.3390/e22040477_

Round 1

Reviewer 1 Report

    This paper proposed a combination of analytical and numerical techniques to estimate higher-order cumulants of the firing probability distributions.

   The results obtained by the authors are interesting and the numerical supported the results effectively. In addition, the paper is well organized and written except several formats and spellings. The followings should be modified:

1,Line 21, is → are,

2,Line 80, a → an,

3, Line 95, no space,

4, Line 99, no space,

5, Line 103, no space,

6, Line 107, no space,

7, Line 120, no space,

8, Line 222, no space.

Author Response

We much appreciate the comments and summary of our manuscript made by the Reviewers. We also would like to thank their exhaustive analysis of our work. We have modified the text to address all the comments of the reviewers, as requested. 

Reviewer 2 Report

Higher-order cumulants drive neuronal activity patterns, inducing UP-DOWN states in neural populations

The authors described a combination of analytical and numerical techniques to estimate higher-order, above the second, cumulants of the firing probability distributions.  They found using the Binder cumulant, that higher-order correlations in the outputs generate a critical neural system that portrays a second-order phase transition.

In my opinion, this manuscript has these good points:

  • The subject is interesting and might absorb many readers in the field;
  • Adequately written and nicely presents the idea;
  • Adequate analytical representations;

Also, there are some suggestions that would increase the strength of the paper which is listed bellows;

  • One of my major points about your article despite interesting topic concerns the novelty of this article, authors should highlight their contributions.
  • I would strongly recommend considering a review discussion regarding such methods and try to show how much the proposed approach is far from these approaches.
  • Please provide a more detailed connection with Shannon entropy and Fisher information.

Some editorial points:

  • Please enhance the quality of your Figures and Tables;
  • Please provide an editable figure in case that they cannot be seen, one can track the context (if possible).

In my opinion, this work is very nice to be published after some slight improvements.

Thank you

Round 2

Reviewer 1 Report

After the modification of the formats I have pointed last time, I recommend it to be accepted. 

Reviewer 2 Report

The manuscript has been improved considerably. Therefore, I accept this manuscript for publishing.